# The Quality of Entrepreneurial Activity and Economic Competitiveness in European Union Countries: A Panel Data Approach

**Valentina Diana Rusu** [1,*] and **Adina Dornean** [2]

[1] Institute of Interdisciplinary Research, Social Sciences and Humanities Research Department, Alexandru Ioan Cuza university of Iași, Lascăr Catargi Street no. 54, 700107 Iași, Romania

[2] Department of Finance, Money and Public Administration, Faculty of Economics and Business Administration, Alexandru Ioan Cuza University of Iași, Carol I Boulevard, No. 22, 700505 Iași, Romania; amartin@uaic.ro

[*] Correspondence: valentinadiana.ig@gmail.com; Tel.: +40-074-297-3287

**Abstract:** To increase competitiveness, a country has to outperform its competitors in terms of research and innovation, entrepreneurship, competition, and education. In this paper, we aim to test the relationship between the quality of entrepreneurial activity and the economic competitiveness for the European Union countries by using panel data estimation techniques. Our research considers a sample of 28 EU countries over the period 2011–2017. For the empirical investigation we apply panel data regression models. The results obtained show that business, macroeconomic environment and the quality of entrepreneurship are significant determinants of economic competitiveness of EU countries. Thus, we identify significant positive relations between innovation rate, inflation rate, FDI and economic competitiveness, and significant negative relations between expectations regarding job creation, tax rate, costs and competitiveness. Our study completes the literature by analyzing the relationship between the quality of entrepreneurship and the competitiveness of countries, for an extensive sample formed by all the 28 countries members of the European Union for a period of seven recent years.

**Keywords:** entrepreneurship; global competitiveness index; innovation; job creation; panel data

## 1. Introduction

The role of entrepreneurship for ensuring economic growth and development of countries has been highlighted in the literature (Wennekers and Thurik 1999; Toma et al. 2014; Meyer and Meyer 2017; Meyer and de Jongh 2018). These studies point out the role played by the development of small and medium enterprises sector for improving economic and social outlooks. Moreover, the role of innovative entrepreneurship is even greater for stimulating the economic development of countries (Bashir and Akhtar 2016; Doğan 2016). Countries with higher levels of innovative entrepreneurs are benefiting from higher levels of economic development.

Therefore, in the European countries the interest of policy makers in increasing national and regional competitiveness determines them to adopt different measures to support a more qualitative entrepreneurship.

Starting from those stated above, the main research question to be analyzed in this paper is how the quality of entrepreneurial activity measured by innovation and job creation can play a role in the promotion of national competitiveness. This is an important topic because, as presented above, countries are increasingly competing with each other and the improvement of competitiveness is seen as a way to sustain economic growth and development. Additionally, the influence of qualitative

entrepreneurship, measured by innovation and job creation, on national competitiveness has been poorly discussed in the literature, so our paper aims at filling this gap by realizing an extensive study for all 28 member countries of the European Union, on a period of seven recent years, 2011–2017.

For testing our hypotheses, we use panel data estimation techniques choosing as a dependent variable of the econometric models the economic competitiveness of countries. As explanatory variables, we took into account three groups of indicators. The first a set comprises of indicators expressing the quality of entrepreneurial activity, namely: Innovation rate measured as percentage of Total early stage entrepreneurial activity and high job creation expectation rate measured as percentage of Total early stage entrepreneurial activity. The other two sets of indicators are used as control variables and comprise of a set of macroeconomic variables: Gross Domestic Product (GDP) growth, inflation rate, tax rate and foreign direct investments (FDI), and business environment indicators: The cost of business start-up procedures.

Our study contributes to the literature by considering for the analysis all the 28 European Union member countries, in comparison with other studies that have analyzed only one or several countries. Another novelty of our analysis is the fact that we have analyzed the relationship between the quality of entrepreneurship and national competitiveness. In the literature are only few studies that analyze the impact of several aspects of entrepreneurship on the level of economic competitiveness of countries, but, to our knowledge, there are no studies testing the relation between the quality of entrepreneurial activities and national competitiveness. Therefore, we intended to fill this literature gap. Moreover, we consider that our results could be of interest to policy makers from the European countries interested to enhance national competitiveness, because it points out the key role played by qualitative entrepreneurship, triggered by the innovative ideas and the creation of new jobs, for higher competitiveness.

This paper is structured as follows. The next section reviews the literature regarding the quality of entrepreneurship measured by innovation and job creation and economic competitiveness and reviews the main empirical studies integrating the relationship between entrepreneurial activity and competitiveness. Section three presents the methodology used for our empirical investigation, describing the sample, the variables considered and the methods used for analyzing the data. Section four summarizes the results obtained and several discussions regarding the results. The final section presents the conclusions and some future directions of research.

## 2. Literature Review

In the economic and business literature, there is no consensus regarding the concept of economic competitiveness. Even so, it is considered a very complex concept (Rusu and Roman 2018) which refers to "the favourable position of a country, especially in international trade, but also the ability to improve its position".

We have found several approaches of the competitiveness according to different organizations. For instance, Organization for Economic Cooperation and Development (OECD) defines competitiveness by taking into account its two main reference levels—the firm and the nation. For our research it is important to clarify the concept of national competitiveness. Thus, according to OECD and to the fact that the objective of competitiveness for a nation is to maintain and improve its citizens' living standards, competitiveness is considered to be "the ability of companies, industries, regions, nations or supranational regions to generate, while being and remaining exposed to international competition, relatively high factor income and factor employment levels on a sustainable basis" (Hatzichronoglou 1996).

A similar approach has the Institute for Management Development (IMD) which refers to competitiveness both as a tool and an objective of economic policy. Arturo Bris (IMD World Competitiveness Centre 2018a), the director of the IMD World Competitiveness Centre understands competitiveness as "the ability of countries, regions and companies to manage their competencies to achieve long-term growth, generate jobs and increase welfare". Also, when defining economy's competitiveness, "it cannot be reduced only to GDP and productivity" because enterprises also

have to deal with political, social and cultural dimensions. Thus, governments need to provide "an environment characterized by efficient infrastructures, institutions and policies that encourage sustainable value creation by the enterprises" (IMD World Competitiveness Centre 2018b).

Our study is focused on European Union countries, so it is important to mention the position of European Commission regarding the economic competitiveness. European Competitiveness Report (European Commission 2014) considers a competitive economy as being that economy that has a consistently high rate of productivity growth. The Europe 2020 Strategy describes the seven pillars of competitiveness (enterprise environment, digital agenda, innovative Europe, education and training, labor market and employment, social inclusion and environmental sustainability) which has been combined in order to create the Europe 2020 Competitiveness Index (World Economic Forum 2012). In their research, Radulescu et al. (2018) investigated the implementation of Europe 2020 Strategy for six selected CEE countries (Romania, Bulgaria, the Czech Republic, Poland, Hungary and Slovakia) over the period 2004–2015, considering that fulfilling the Strategy objectives enhance the economic performance and competitiveness. Performing the study, they highlighted that the most important factor which contributes to economic performance and competitiveness is represented by the tertiary level of education, followed by the school dropout ratio, the share of renewable energy in final energy consumption, and the employment rate.

On the other hand, The World Economic Forum has been studying Europe's competitiveness compared with that of the United Stated, beginning with 1979. According to the World Economic Forum (2014), "Competitive economies are those that are able to provide high and rising living standards, allowing all members of a society to contribute to and benefit from these levels of prosperity. In addition, competitive economies are those that are sustainable—meeting the needs of the present generation while maintaining the ability to meet those of future generations".

Considering the definitions mentioned above, we can assert that competitiveness is a complex concept. All the definitions have something in common: Economic and sustainable growth in the context of a favourable business environment.

The objective of our paper is not to define the economic competitiveness, but to link the economic competitiveness of nations to entrepreneurial activity. Studies in the field showed that in the context of globalization, the global business environment, innovation and creativity are considered key ingredients in creating and sustaining economic competitiveness (Ojo et al. 2017; Baron and Tang 2011). A similar perspective is shared by Anastassopoulos (2007), who considers that the enterprises and the environment in which they operate are important determinants of economic competitiveness. In this context, a competitive strategy and performance is necessary to be defined and applied.

The quality of entrepreneurship is very important for the development of an economy, and the innovative entrepreneurs are seen as agents helping markets development and the increase of economic competitiveness. As shown by Bosma et al. (2012), the improvement of entrepreneurial environment of a country could be a key factor for increasing economic competitiveness. In this context, researchers tried to define and to measure the relationship between economic development and entrepreneurship because entrepreneurship has been recognized as a micro driver of innovation and economic growth (Wennekers et al. 2010; Hall et al. 2010; González-Pernía et al. 2015; Bashir and Akhtar 2016; Chowdhury et al. 2018). In their study, González-Pernía et al. (2015) highlighted the importance of innovative entrepreneurs. Even if they represent a small portion of the entire population of business founders, they have an extraordinary economic impact, as they develop new technologies, create new jobs and enhance the revitalization capacity of territories.

Grilo and Thurik (2005) consider that the entrepreneurial activity is at the heart of innovation, productivity growth, competitiveness, economic growth and job creation and this explain why entrepreneurship has become a key policy issue (Wennekers and Thurik 1999) and why policymakers have to take into consideration the relationship between entrepreneurship and economic development.

Recent studies (Amorós et al. 2013) reveal that entrepreneurship is very important for a country's competitiveness and development because entrepreneurs create new businesses and in turn these

generate new jobs, more competition, and may even increase productivity through innovation. The same opinion is shared by Gonzalez-Sanchez (2013) who considers that innovation and entrepreneurial activities have become increasingly important elements for economic growth and are also decisive factors in a country's level of development. In their study on European countries they have found that the effects of innovation and entrepreneurial activity tend to be more positive for the economy of those countries when the economic scenario worsened.

A study (2001) applied to the province of Seville, one of the least developed areas in the European Union in the 2000's, showed that entrepreneurship and the entrepreneurs who innovate are important for economic development.

To understand m Guzmán and Santos (2001) ore deeply how entrepreneurship can make its contribution to economic development, Pawitan et al. (2017) analyzed the relationship between entrepreneurial spirit, a subset of entrepreneurship, and global competitiveness at the national level for the case of Indonesia in 2015. The authors understand the term of entrepreneurial spirit to consist of two dimensions: Entrepreneurial attitudes (social value, personal attributes, and goal orientation) and entrepreneurial activities (total early entrepreneurial activities and rate of established business ownerships). Their results indicate that global competitiveness can be improved through personal attributes and goal orientation while the indicators of entrepreneurial activities are negatively correlated with global competitiveness.

In their empirical study, Bashir and Akhtar (2016) conducted a survey of more than 1500 entrepreneurs across the G20 countries in order to explore the relation of Innovative Entrepreneurship and economic growth and its role in economic development of G20 member countries. Their results show a positive relationship which demonstrates that it is possible to increase economic growth through innovative entrepreneurship.

In addition, several studies have investigated the impact of entrepreneurship on countries' economic and competitiveness development. A recent one (Dhahri and Omri 2018) investigated the relationship between entrepreneurship and the three areas of sustainable development (economic, social and environmental) for the case of 20 developing countries (Argentina, Brazil, China, Colombia, Egypt, India, Indonesia, Iran, Malaysia, Mexico, Morocco, Nigeria, Pakistan, Peru, Philippines, Romania, South Africa, Thailand, Tunisia and Turkey) over the period 2001–2012. Dhahri and Omri (2018) provide results that confirm the positive contribution of entrepreneurship to the economic and social dimensions of sustainable development, while its contribution to the environmental dimension is negative.

In a different approach, Bosma et al. (2018) pointed out the impact of institutions on "productive entrepreneurship" and the effects of entrepreneurship on economic growth. The authors (Bosma et al. 2018) use the definition of Baumol (1993) for the concept "productive entrepreneurship" which refers to "any entrepreneurial activity that contributes directly or indirectly to net output of the economy or to the capacity to produce additional output". This study is important because productive entrepreneurship includes entrepreneurship that generates innovation and economic growth and the results showed the contribution of productive entrepreneurship to economic growth for a sample of 25 European countries over the analyzed period 2003–2014. Thus, their study confirms also that innovation, as a channel of entrepreneurship may drive economies to economic growth.

European Union policy regards innovation as an important driver for the firms' competitiveness, economic growth and job creation (European Commission 2014), aspect emphasized and demonstrated also by various research studies. Thus, we mention the research of Ciocanel and Pavelescu (2015) who analyzed the link between innovation and economic competitiveness in the EU context (EU countries and Norway) over the period 2008–2013. Their results concluded that improving of innovation performance leads to the increasing of national competitiveness. On the other hand, the correlation between innovation and economic growth, and, implicitly, competitiveness has been studied by Petrariu et al. (2013) for Central and Eastern European (CEE) countries. Even innovation is often considered to be a typical activity of the developed countries, Petrariu et al. (2013) showed for the

group of CEE countries that innovation makes significant contribution to national competitiveness and economic growth. They also argue that the gap between the developed (Western) and developing (Eastern) economies can be reduced by investing in innovation.

Regarding the developed economies of the European Union, the research of Bartz and Winkler (2016) on German businesses tested if financial instability, such as 2009 crisis, is detrimental to entrepreneurship. Their results highlighted that entrepreneurial activity is riskier during crisis than in normal times, but the interesting fact is that small firms exhibit a relative growth advantage compared to larger firms in both stable and crisis times, and this is considered to be a flexibility advantage of small size firms. Developing economies and the role of innovation in stimulating competitiveness and economic growth was the subject of another recent paper (Terzić 2017). The study comprised 10 developing countries from European Union in order to determine the interconnections between the variables of innovation, competitiveness and growth. The results obtained showed that for the selected countries, innovation performance depends on a developed research system, improved conditions for entrepreneurship, and a higher degree of innovation performances. Through job creation and the development of new products and services, innovation contributes to the increase of competitiveness and represents a key factor leading countries' economic growth (Kuhlman et al. 2017). Innovation represents an important pillar for global competitiveness (Ghoniem and El Khouly 2012) that contributes also to the improvement of international competitiveness as found by Özçelik and Taymaz (2004) in their study on Turkey. Thus, public authorities should apply appropriate policies and plans for actions that increase innovation which will enhance economic growth. Potluka and Dvoulety (2018) have emphasized that the policy makers from Czech Republic actively support companies from public budgets in order to sustain national competitiveness and to ensure higher levels of employment. Czechpublic programs are intended to support innovative companies and thus, increase competitiveness and employment. The authors have found a significant positive impact of the public programmes on employment, sales and profit.

Regarding emerging economies, there is limited research on entrepreneurship, and we cannot apply the findings about the world's developed economies to them. Taking into consideration the fact that entrepreneurship plays a key role in the economic development, Bruton et al. (2008) suggest there is a strong need to develop an understanding of entrepreneurship in emerging economies.

Regarding European Union countries, the authors who have analyzed the relationship between entrepreneurship and national competitiveness (Bosma et al. 2018; Ciocanel and Pavelescu 2015; Szabo and Herman 2012) found also that innovative entrepreneurs contribute to economic growth and enhance economic competitiveness both in developed countries and in developing countries of the EU.

Bulat et al. (2018) conducted a comparative study on 50 member countries of the Eurasian Economic Union in order to identify the factors that most significantly affect the competitiveness of their economy. Using a multiple regression model, Bulat et al. (2018) demonstrated that the innovation index has a significant impact on the competitiveness of the economy. Starting from these results, the authors proposed several measures for the economy of Kazakhstan, measures designed to stimulate innovation, which can also help increase competitiveness.

A table representation of the literature analyzing the most important aspects related with our research is presented below in Table 1.

**Table 1.** Authors included in the literature review and their findings.

| Findings | Authors |
| --- | --- |
| Innovative entrepreneurship increases economic growth and ensure economic development | Porter 1990; Romer 1994; Wennekers and Thurik 1999; Johansson et al. 2001; Szabo and Herman 2012; Toma et al. 2014; Kritikos 2014; Bashir and Akhtar 2016; Doğan 2016; Meyer and Meyer 2017; Meyer and de Jongh 2018; Bosma et al. 2018 |
| Positive link between innovation and creativity and economic competitiveness | Anastassopoulos 2007; Baron and Tang 2011; Bosma et al. 2012; Ojo et al. 2017 |
| Entrepreneurship as a micro driver of innovation and economic growth | Wennekers et al. 2010; Hall et al. 2010; González-Pernía et al. 2015; Bashir and Akhtar 2016; Chowdhury et al. 2018 |
| Innovation and entrepreneurial activities determine economic growth, | Amorós et al. 2013; Gonzalez-Sanchez 2013; Dhahri and Omri 2018 |
| Positive link between innovation and economic competitiveness | Cantwell 2003; Özçelik and Taymaz 2004; Ghoniem and El Khouly 2012; Petrariu et al. 2013; Kritikos 2014; Huggins et al. 2014; Ciocanel and Pavelescu 2015; Terzić 2017; Matos Ferreira et al. 2017; Herman 2018; Bulat et al. 2018 |
| Explanatory variables | |
| Entrepreneurial activities have positive effects on employment in short and long term | Kritikos 2014 |
| Significant direct relationship between innovation of entrepreneurial activities and competitiveness | Cantwell 2003; Özçelik and Taymaz 2004; Ghoniem and El Khouly 2012; Petrariu et al. 2013; Huggins et al. 2014; Ciocanel and Pavelescu 2015; Doğan 2016; Matos Ferreira et al. 2017; Herman 2018 |
| Positive correlation between employment and increase in competitiveness | Moser et al. 2010; Martus 2013; Rusek 2015; OECD 2016; World Bank Group 2017 |
| Negative relationship between high job creation expectation rate and economic competitiveness | Chen et al. 2007; Gallegati et al. 2014; Rusek 2015; Semmler and Chen 2017; World Bank Group 2017 |
| Significant positive relationship between economic growth and national competitiveness | Podobnik et al. 2012; Dobrinsky and Havlik 2014; Korez-Vide and Tominc 2016 |
| Control variables | |
| *control variable—tax rate* positive relationship between tax rate and national competitiveness negative relationship between tax rate and national competitiveness | Miller and Kim 2008; Knoll 2010 Summers 1988; Gray and Holtz-Eakin 2009; Ecorys 2014 |
| *control variable—inflation rate* positive relationship between inflation rate and national competitiveness negative relationship between inflation rate and national competitiveness | Vidal-Suñé and Lopez-Panisello 2013; Sayed and Slimane 2014; Rusu and Roman 2018 Salman 2014; Iarossi 2009 |
| *control variable—foreign direct investments (FDI)* positive relationship between FDI and national competitiveness | Meyer and Sinani 2009; Kim and Li 2014 Fontagné and Pajot 1997; Javorcik 2004; Ocharo and Musyoka 2018; Domazet and Marjanović 2018 |
| *control variable—cost of business start-up procedures* negative relationship between cost of business start-up procedures and economic competitiveness | Iarossi 2009; Globerman and Georgopoulos 2012; Messaoud and El Ghak Teheni 2014 |

Source: authors findings.

Entrepreneurial activity and competitiveness have been the research topic of a large number of authors, enjoying increased attention on how entrepreneurs innovate and consequently contribute to higher levels of competitiveness. Reviewing the literature, we can assert that the quality of entrepreneurship, measured by innovation and job creation, is important in order to achieve economic competitiveness.

## 3. Materials and Methods

The main objective of our investigation is to determine the impact of the quality of entrepreneurship on host countries' competitiveness. In order to achieve this goal, we consider the innovation rate of entrepreneurial activity and the potential creation of new jobs in the future as explanatory variables, besides some control variables, such as the level of economic growth, inflation, total tax rate, foreign direct investments and the costs of business start-up procedures. For measuring the level of national competitiveness, we use as proxy the Global Competitiveness Index.

The sample comprises information for the 28 European Union member countries between 2011 and 2017. We have chosen this period for analysis due to the availability of data. This period includes the years after the recent financial crisis and is marked by recession and recovery efforts from European economies. The effects of the recent financial crisis were felt with different intensities in different countries, so, we must specify that our results might be influenced by this situation. If we would have analyzed only years without difficulties in the EU economies, we might have obtained different results. But, as the economy is generally marked by cyclicality between positive phases and recession or crisis, we cannot make a separation, and we consider that is important to analyze the relationship between the quality of entrepreneurship and national competitiveness also in difficult economic times.

The general equation of our econometric model is described below by Equation (1):

$$GCI_{it} = \beta_0 + \beta_1\, X_{it} + \beta_2\, Y_{it} + \alpha_i + \varepsilon_{it} \tag{1}$$

where $i$ represents the EU countries ($i = 1, \ldots, 28$), and $t$ represents time ($t = 2011, \ldots, 2017$). $GCI_{it}$ is the dependent variable and represents the Global Competitiveness Index calculated by the World Economic Forum. $\beta_0$ is the common intercept and $\beta$ is the vector of coefficients associated with the explanatory variables. $X_{it}$ is the vector of explanatory variables for country $i$ at time $t$. $Y_{it}$ is the vector of explanatory variables for country $i$ at time $t$. $\varepsilon_{it}$ is the random term for country $i$ at time $t$.

The general equation adapted to our sample is described in the following by Equation (2):

$$GCI_{it} = \beta_0 + \beta_1\, \text{innov}_{it} + \beta_2 jobs_{it} + \beta_3 Macroec_{it} + \beta_3 business_{it} + \alpha_i + \varepsilon_{it} \tag{2}$$

Since our sample combines time series and cross-sections, we will apply a regression model on a balanced panel data for analyzing the effects of the considered explanatory variables on competitiveness of EU member countries. In order to study a model with these characteristics, we can use two different models: A fixed effect model (FE) and a random effect model (RE). Fixed effect model explores the relationship between the predictor and outcome variables within an entity and assumes that the independent variables are fixed across observation units and that the fixed effects are computed from the differences within each unit across time. Differently, Random effect model is usually preferred when we think that there are no omitted variables or if we believe that the omitted variables are not correlated with the explanatory variables considered in the model. Using this model will determine unbiased estimates of the coefficients, use all the data available, and produce the smallest standard errors. The significant distinction between fixed and random effects is whether the unobserved individual effect incorporates elements that are correlated with the regressors in the model (Greene 2003; Baltagi 2005). For choosing between OLS, FE and RE we apply Hausman test and Redundant fixed effects test.

Also, when running the panel data regression models we determined the estimator variance–covariance matrix by the White cross method, treating the pool regression as a multivariate regression, to cope with the suspicion of transverse heteroscedasticity.

As previously mentioned, we consider as dependent variable of our model the competitiveness of European Union member countries. Since in the literature there is not an unanimous definition of the competitiveness of a country we decided to measure it by the Global Competitiveness Index (GCI), which is calculated by the World Economic Forum (WEF). WEF calculates GCI by taking into account twelve pillars, which are grouped into three sub-indexes. The first sub-index refers to basic requirements and comprise of institutions, infrastructure, macroeconomic environment and health and primary education. The second sub-index refers to efficiency enhancers and is considering higher education and training, goods and labor market efficiency, financial market development, technological readiness and market size. The last sub-index consists of two pillars, namely: Business sophistication and innovation. The GCI takes values from 1 to 7, if its value is higher it means that the country has a higher level of competitiveness.

Analysing the average GCI for the European Union (see Figure 1) we observe that it had an increasing trend in the analyzed period, with a break-out point in 2013. The decrease in 2013 can be explained by a slowdown of European economic growth. In 2014, the competitiveness of EU countries registered a significant improvement, and continued the increasing trend in the following years, as a result of the new measures taken by the EU through the Europe 2020 Strategy in order to support entrepreneurship and national competitiveness (European Commission 2010). The Entrepreneurship 2020 Action Plan has the purpose to sustain entrepreneurial potential of the European citizens, and to remove existing obstacles and revolutionize the culture of entrepreneurship in the EU. This action plan intends to ease the creation of new businesses and develop a business environment much more supportive for existing entrepreneurs in order to simulate them to grow and be more innovative (European Commission 2013).

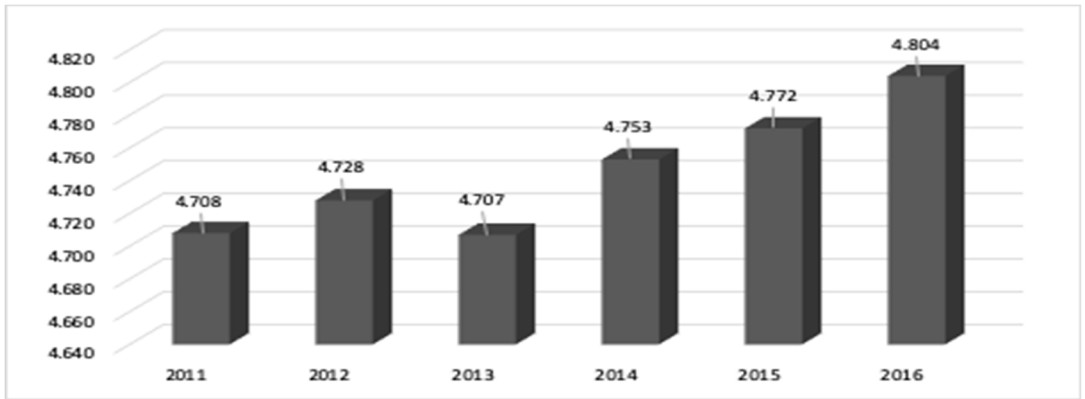

**Figure 1.** The dynamics of average GCI for EU-28 countries. Source: own calculation.

The set of explanatory variables encompasses innovation rate and high jobs creation expectation rate, as well as several control variables. The variables, their measurement and their source are presented below in Table 2. We considered the innovation rate as an explanatory variable that can help measuring the quality of entrepreneurial activities. As shown by Kritikos (2014) radical innovations often lead to economic growth and the entrepreneurs who bring innovations to the market offer a significant contribution to economic progress. The author also highlight that innovative entrepreneurs are vital to the competitiveness of the economy. However, the gains of entrepreneurship are realized only if the business environment is receptive to innovation.

**Table 2.** The variables of the model, their measurement and source.

| Variable | Measurement | Source |
|---|---|---|
| Economic competitiveness of countries (GCI) | The Global Competitiveness Index which takes scores from 1 to 7 (a higher average score means a higher degree of competitiveness) and is calculated as a weighted average of several components of competitiveness that are grouped into 12 pillars. | World Economic Forum |
| Innovation rate (INNOV) | Percentage of those involved in Total early stage entrepreneurial activity who indicate that their product or service is new to at least some customers and that few/no businesses offer the same product. Total early stage entrepreneurial activity represents the percentage of the population with the age between 18–64 who are either a nascent entrepreneur or owner-manager of a new business. | Global entrepreneurship Monitor |
| High job creation expectation rate (HJOB) | Percentage of those involved in Total early stage entrepreneurial activity who expect to create 6 or more jobs in 5 years. | Global entrepreneurship Monitor |
| *Control variables* | | |
| GDP growth (GDP) | Annual percentage growth rate of GDP at market prices based on constant local currency. | World Bank DataBank |
| Inflation rate (INFL) | Measured by the annual growth rate of the GDP implicit deflator shows the rate of price change in the economy as a whole. | World Bank DataBank |
| Tax rate (TAX) | Total tax rate measures the amount of taxes and mandatory contributions payable by businesses after accounting for allowable deductions and exemptions as a share of commercial profits. | World Bank DataBank |
| Foreign direct investments inflows (FDI) | Net inflows (new investment inflows less disinvestment) in the reporting economy from foreign investors and is divided by GDP. | World Bank DataBank |
| Cost of business start-up procedures (COST) | The costs supported when starting a new business measured as percentage of GNI per capita. | World Bank DataBank |

Source: Own elaboration.

In addition, the mentioned study has shown that entrepreneurial activities have positive effects on employment on a short and long term, but negative effects on a medium term. These relations were obtained for the US, for a number of European countries, and for 23 OECD countries and are explained by the fact that in the incipient stage a significant positive effect on employment appears because the newly created firms will generate new jobs. However, after the business passes the initial phase,

usually a stagnation phase or even a negative effect on employment is noted because, in this phase, the new businesses are gaining market share from the companies that cannot compete and because a part of the new firms fail. After this middle phase, the increased competitiveness of suppliers leads to positive gains in employment once again. About ten years after start-up, the impact of new business formation on employment has finally faded away (Kritikos 2014).

Several others studies (Cantwell 2003; Özçelik and Taymaz 2004; Ghoniem and El Khouly 2012; Petrariu et al. 2013; Huggins et al. 2014; Ciocanel and Pavelescu 2015; Doğan 2016; Matos Ferreira et al. 2017; Herman 2018) also emphasize the significant direct relation between innovation of entrepreneurial activities and competitiveness, and describe the significant role played by the entrepreneurs that find new methods of production, create new products or better products at lower costs or use new ways of organizing their activity for the increase of national competitiveness. With the help of innovative activities, the entrepreneurs will obtain a competitive advantage and maintaining this advantage correlated with continuous development will determine the increase of national competitiveness. Therefore, innovation is seen as the motor that drives the progress of competitiveness and economic development of a country (Johansson et al. 2001; Romer 1994), and the competitiveness of a country depends on the capacity of its industry to apply innovation and increase its quality (Porter 1990). The study of Szabo and Herman (2012) also pointed out the need to increase innovation rate in order to enhance competitiveness especially in emerging countries of European Union. Nevertheless, we have to keep in mind also the fact that increased national competition creates pressure on entrepreneurs to be innovative.

For measuring the quality of entrepreneurial activities from the European Union countries we also have considered another variable: high job creation expectation rate. As argued by the European Commission (1994) competitiveness, growth and employment are closely interrelated. The white paper of the European Commission (1994) shows that increasing employment threshold means increasing the overall productivity of a country which will guarantee an increase of the international competitiveness of the country. The employment threshold represents the percentage change above which the growth rate of GDP leads to an increase in employment. Moreover, Regional competitiveness and employment was one of the three priority objectives of the European Union under European cohesion policy for 2007–13, and it aimed to strengthen the competitiveness and increase employment in the regions that were not included in the Convergence Objective. Among its main purposes it was the promotion of innovation and sustaining entrepreneurship in relation to the increase of regional competitiveness. In the current 2014–20 programming period, most of the regions previously covered by the Regional Competitiveness and Employment Objective receive funding in their quality as more developed regions or transition regions (European Commission Web Site 2019).

Other papers (Moser et al. 2010; Martus 2013; Rusek 2015; OECD 2016; World Bank Group 2017) also highlight the importance of job creation for ensuring economic growth and increased national competitiveness, identifying a positive correlation between employment and increase in competitiveness, although, in some countries this effect is small. Also, these papers emphasize the importance of supporting entrepreneurship and SMEs in order to promote growth and strengthening the local economic base. Martus (2013) pointed out that rising unemployment rates decrease regional living standards and competitiveness and is underlining the importance of keeping high rates of employment for ensuring regional competitiveness. The author also highlights the mutual determination relationship between employment and competitiveness, as ensuring low unemployment rates will increase competitiveness and higher level of national competitiveness are related with lower unemployment rate.

However, the World Bank report (World Bank Group 2017) points out that the interaction between competitiveness, expressed through productivity, and jobs is both conceptually and empirically more complex and depends both on the context and time of analysis. Thus, stimulating competitiveness through industrial upgrading can also induce dislocations as resources shift within and between sectors, which can determine unemployment and can destabilize some firms, industries, or whole regions. Moreover, the effort to restore and improve the competitiveness implies the improvement

of the unit labor costs which in turn requires a higher productivity (Rusek 2015) but productivity creates unemployment on short and medium terms, and employment in the long run (Chen et al. 2007; Gallegati et al. 2014; Semmler and Chen 2017).

Starting from those stated above, we formulate several hypotheses. Our main hypothesis (H1) is that the quality of entrepreneurship is a significant factor that is directly influencing the national competitiveness. When the quality of entrepreneurial activity is rising will determine an increase of national competitiveness.

Also, starting from the two explanatory variables considered in our empirical analysis we formulate two sub-hypotheses, namely:

**Hypothesis 1.1 (H1.1).** *Innovation rate has a direct and positive impact on economic competitiveness of EU member countries.*

**Hypothesis 1.2 (H1.2).** *High job creation expectation rate has a direct and positive impact on economic competitiveness of EU countries.*

Besides the explanatory variables we also use several control variables for ensuring the robustness of our results. These control variables are measuring macroeconomic and business environment conditions and, as shown by previous studies, might influence both entrepreneurship (Grilo and Thurik 2004; Hoffmann et al. 2006; Vidal-Suñé and Lopez-Panisello 2013; Aparicio et al. 2016; Roman et al. 2017) and national competitiveness (Miller and Kim 2008; Knoll 2010; Podobnik et al. 2012; Vidal-Suñé and Lopez-Panisello 2013; Sayed and Slimane 2014; Dobrinsky and Havlik 2014; Korez-Vide and Tominc 2016; Rusu and Roman 2018). As highlighted by the mentioned studies, rich countries that have lower business regulations and higher inflows of foreign direct investments are more competitive than poor countries, with high level of tax rate, more regulations and a deficit of foreign investments.

Several studies (Podobnik et al. 2012; Dobrinsky and Havlik 2014; Korez-Vide and Tominc 2016) have emphasized the significant positive relationship between economic growth and national competitiveness showing that countries with higher levels of economic growth are more competitive. Also, higher economy growth has positive effects on entrepreneurship by creating new business opportunities.

The tax rate plays also a significant role for the national competitiveness because excessive tax burdens are considered to be responsible for the poor international performance of industries and high corporate tax rates are considered to undermine the international competitiveness of a country. The reduction of the corporate tax rates could be a way for attracting more investment capital and could increase firms' productivity and investment incentives (Miller and Kim 2008; Knoll 2010). Contrariwise Summers (1988), those tax measures which might stimulate the attraction of funds from abroad can determine an appreciation in the real exchange rate and at the same time a reduction in the international competitiveness of national industries. Therefore, the relationship between tax rate and national competitiveness could be either negative or positive.

The relationship between inflation rate and competitiveness can also be analyzed from two points of view. An increase of inflation can determine an improvement of business opportunities explained by the fact that higher price levels can increase earnings expectations of entrepreneurs and can stimulated business development and implicitly enhance competitiveness (Vidal-Suñé and Lopez-Panisello 2013; Sayed and Slimane 2014). However, increased inflation also increases the costs for business start-up and activity and might affect negatively the entrepreneurs (Salman 2014). Regulations about doing business, often expressed by the higher level of costs for starting and running a business are negatively influencing the entrepreneurial activity and lower competitiveness (Iarossi 2009).

Foreign direct investments (FDI) stimulate the national competitiveness by the fact that inflows of foreign capital increase employment, offer more funds for the businesses and stimulates them to become innovative, determine the development of national industries and stimulate exports of goods.

The positive effect of FDI on national competitiveness and entrepreneurship depends on the level of development of the country (Meyer and Sinani 2009; Kim and Li 2014).

Therefore, according to the empirical results from the literature the growth of GDP, inflation rate, tax rate, foreign direct investments inflows and costs for starting a new business are significantly influencing national competitiveness.

Starting from the aforementioned theoretical aspects in this paper we will address the following research model (see Figure 2).

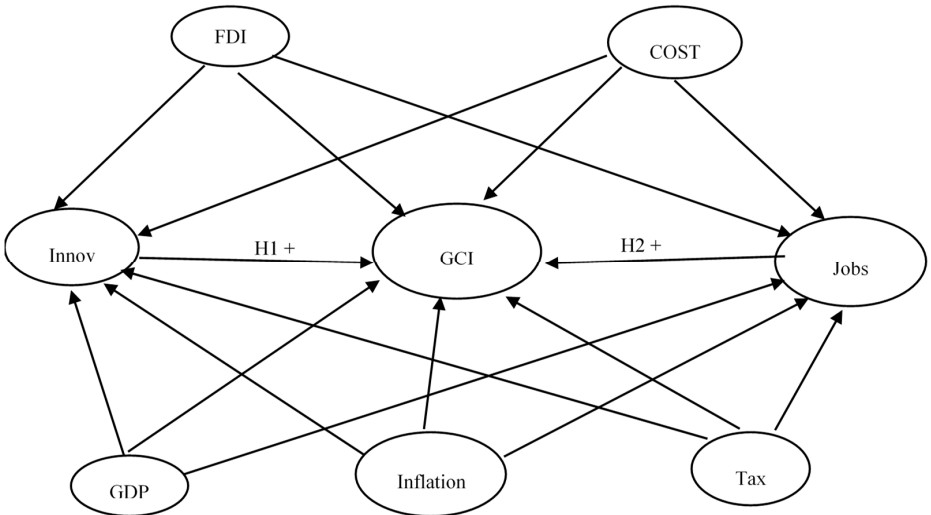

**Figure 2.** Research model with hypotheses.

## 4. Results and Discussions

The important descriptive statistics of global competitiveness index and of the independent variables are provided in Table 3. The number of observations obtained for the variables used for our model was different from country to country because of the lack of available data during the analyzed period of time. For some of the variables we have obtained a different number of observations because, for some countries were not available data for all the years considered in the analysis. The summary of the descriptive statistics emphasizes the fact that the GCI data are distributed between a minimum level of 3.85 (in Greece, 2012) and a maximum of 5.61 (in Sweden, 2011). The value of standard deviation shows relatively small variations of this index between the EU countries and for the analyzed period.

**Table 3.** Descriptive statistics of dependent and explanatory variables.

|  | Mean | Median | Maximum | Minimum | Std. Dev. | Obs. |
|---|---|---|---|---|---|---|
| GCI | 4.739859 | 4.546944 | 5.612262 | 3.859568 | 0.504846 | 162 |
| INNOV | 29.73850 | 28.90000 | 54.10000 | 8.620000 | 7.842005 | 133 |
| JOBS | 22.80083 | 21.30000 | 48.40000 | 4.300000 | 9.301215 | 133 |
| GDP | 1.774781 | 1.840155 | 25.55727 | −9.132.494 | 3.128404 | 162 |
| INFL | 1.338962 | 1.253974 | 7.280033 | −2.352.300 | 1.442996 | 162 |
| TAX | 41.82778 | 42.35000 | 70.80000 | 18.40000 | 12.93358 | 162 |
| FDI | 7.714092 | 2.034041 | 252.3081 | −4.346.255 | 27.41299 | 162 |
| COST | 4.288272 | 2.200000 | 20.50000 | 0.000000 | 4.606185 | 162 |

Source: Own calculations.

As regards the explanatory variables, the innovation rate varies between a minimum of 8.62% from TEA (in Bulgaria, 2015) to a maximum of 54.1% (in Denmark, in 2011). The high jobs expectation rate registered a higher variation compared to previous variables, and is distributed between 4.3% (in Greece, in 2015) and 48.4% (Latvia, 2012).

The variations obtained for the control variables show that there are important differences between the EU countries regarding the macroeconomic and business environment indicators. This happens because among the 28 European Union economies, we have countries with different levels of economic development.

In order to obtain accurate results from the empirical analysis, we have also considered the problem of multicollinearity. The correlation test applied to our variables showed that it does not exist multicollinearity between the considered variables, mentioning that we used as the reference point the value of 0.80, similar to other studies (Bryman and Cramer 2001) (see Table 4).

**Table 4.** The correlation matrix of the variables.

|  | GCI | INNOV | JOBS | GDP | INFL | TAX | FDI | COST |
|---|---|---|---|---|---|---|---|---|
| GCI | 1.000 | | | | | | | |
| INNOV | 0.334 (0.000) | 1.000 | | | | | | |
| JOBS | −0.286 (0.000) | 0.100 (0.249) | 1.000 | | | | | |
| GDP | 0.153 (0.076) | 0.255 (0.003) | 0.380 (0.000) | 1.000 | | | | |
| INFL | 0.101 (0.244) | 0.052 (0.551) | 0.374 (0.000) | 0.419 (0.000) | 1.000 | | | |
| TAX | −0.052 (0.546) | −0.178 (0.040) | −0.257 (0.002) | −0.230 (0.007) | −0.008 (0.925) | 1.000 | | |
| FDI | 0.156 (0.071) | 0.132 (0.127) | 0.077 (0.373) | 0.429 (0.000) | 0.105 (0.227) | −0.272 (0.001) | 1.000 | |
| COST | −0.349 (0.000) | −0.269 (0.001) | −0.182 (0.035) | −0.339 (0.000) | −0.114 (0.189) | 0.220 (0.010) | −0.073 (0.399) | 1.000 |

Note: Probability in parenthesis. Source: Own calculations.

Thus, we proceed with the regression analysis. The regression analysis was carried out by applying three different models: Ordinary Least Squares, Fixed effects model and Random effects model. Since we have obtained different results when applying these mentioned models (see Table 5), further we have tested to see which model is better at explaining the relationships identified. Thus, we run two tests to choose between the fixed and random effects: The Hausman test and the redundant fixed effects test (see Tables 6 and 7).

**Table 5.** Effects of entrepreneurship performance on competitiveness.

| Dependent Variable—GCI | PLS | FE | RE |
|---|---|---|---|
| Innovation rate | 0.018 *** (0.005) | 0.016 *** (0.005) | 0.000 (0.001) |
| High job creation expectation rate | −0.024 *** (0.004) | −0.027 *** (0.007) | −0.000 (0.000) |
| GDP growth | 0.004 (0.015) | 0.005 (0.036) | 0.007 *** (0.002) |
| Inflation rate | 0.072 *** (0.030) | 0.077 (0.055) | −0.015 *** (0.003) |
| Tax rate | −0.004 ** (0.001) | 0.004 ** (0.001) | −0.000 (0.001) |
| Foreign direct investments inflows | 0.005 *** (0.001) | 0.005 ** (0.001) | 0.000 (0.000) |
| Cost of business start-up procedures | −0.039 *** (0.003) | −0.040 *** (0.003) | −0.003 * (0.001) |
| Observations | 133 | 133 | 133 |
| R-squared | 0.385 | 0.392 | 0.141 |
| R-squared adjusted | 0.350 | 0.331 | 0.093 |
| F-statistic | 11.197 *** | 6.461 *** | 2.940 *** |

Note: Standard errors in parenthesis. *, ** and *** indicate significance at 10%, 5% and 1% levels, respectively. Source: Own elaboration.

**Table 6.** Results of the Hausman test.

| Correlated Random Effects—Hausman Test | | | |
|---|---|---|---|
| Equation: EQ01 | | | |
| Test Cross-Section Random Effects | | | |
| Test Summary | Chi-Sq. Statistic | Chi-Sq. d.f. | Prob. |
| Cross-section random | 0.000000 | 7 | 1.0000 |

Source: own elaboration.

**Table 7.** Results of the redundant fixed effects test.

| Effects Test | Statistic | d.f. | Prob. |
|---|---|---|---|
| Cross-section F | 246.961942 | (26.95) | 0.0000 |
| Cross-section Chi-square | 562.342712 | 26 | 0.0000 |
| Period F | 3.769796 | (5.95) | 0.0037 |
| Period Chi-square | 24.072461 | 5 | 0.0002 |
| Cross-Section/Period F | 209.838418 | (31.95) | 0.0000 |
| Cross-Section/Period Chi-square | 564.045907 | 31 | 0.0000 |

Source: own elaboration.

Based on the present analysis, the results of the Hausman test indicate that the H0 hypothesis (H0: Random effect is preferred) is strongly accepted (*p* values = 1.000) which means that the random effects model is preferred. The results of the Hausman test are presented in Table 6.

On the other hand, the results obtained for the redundant fixed effects test strongly reject the null hypothesis (H0: The fixed effects are redundant) and indicate that the fixed effects are statistically significant (see Table 7).

Since the results of the two tests show a contradiction, the pooled OLS regression is favoured (Park 2009) and is better fitted for explaining the relations between our variables. The regression analysis was used to test the hypotheses of our empirical study. According to the results, both the considered independent variables measuring the quality of entrepreneurship had significantly influenced the economic competitiveness of EU member countries in 2011–2017. According to the results reported in Table 4, we conclude that the regression model fits the data and the whole model is statistically significant ($R^2$ = 0.38 and *p*-value = 0.00). As shown in the table, adjusted $R^2$ is 0.350 which means that about 35% of global competitiveness variation is explained by the independent variables chosen in the model. In other words, in the EU member countries, global economic competitiveness is influenced by the independent variables used in the model.

As explained in the previous section, the innovation rate is measuring the percentage of individuals involved in entrepreneurial activity who have introduced a new product on the market. Based on the results in table no. 4 (marked in bold), the innovation rate has a positive coefficient and statistically significant (at 1% level), which means the innovation rate is significantly influencing the economic competitiveness of the countries from the European Union. This result indicates that increased performance of the entrepreneurial activity, measured by the creation of new or improved products or processes by the entrepreneurs, is stimulating economic competitiveness of countries, since innovation determines progress and stimulates productivity growth which, in turn, drives prosperity.

This result is consistent with the findings of Cantwell (2003), Özçelik and Taymaz (2004), Ghoniem and El Khouly (2012), Petrariu et al. (2013), Kritikos (2014), Huggins et al. (2014), Ciocanel and Pavelescu (2015), Doğan (2016), Matos Ferreira et al. (2017), Herman (2018) who conclude that innovation is considered as a major force in economic growth and as a key pillar for enhancing global competitiveness.

The variable high job creation expectation rate measures the percentage of entrepreneurs who expect to create 6 or more jobs in 5 years. Our results emphasize that the expectation of creating new

jobs in 5 years as a negative coefficient and statistically significant (at 1% level), which means the creation of new jobs is negatively influencing the economic competitiveness of EU countries. This result is inconsistent with the hypothesis that creation of a higher number of jobs in future years will enhance economic competitiveness.

This inconsistency with the previous studies can be related to the fact that increased employment rates can affect productivity, because the effort to restore and improve the competitiveness implies the improvement of the unit labor costs which in turn requires a higher productivity, but increased productivity creates unemployment on the short and medium terms, and employment in the long term. Also, higher rates of employment imply higher labor costs for the firms which might reduce their profits and the competitiveness, on short term. Moreover, in our case, the potential creation of new jobs is analyzed in relation with higher levels of entrepreneurial innovation, and process innovation is decreasing employment generating thus an inverse relationship between innovation and employment. Moderate levels of innovation can determine a higher natural rate of unemployment, because of increased job turnover from a decline in the length of each job and a time delay between the loss of a job and the acquisition of a new one. However, very high rates of innovation can reduce the natural unemployment rate, producing an inverted 'U' relationship between natural unemployment and innovation rates (Arundel and Kemp 1999).

Our findings are in line with those of Chen et al. (2007), Gallegati et al. (2014), Rusek (2015), Semmler and Chen (2017) and World Bank Group (2017).

Regarding the control variables, our paper is in line with the studies showing that rich countries, with lower regulations and higher level of foreign investments are more competitive than poor countries, with high level of tax rate, more regulations and a deficit of foreign investments. For the coefficient corresponding to inflation rate our findings are similar to those of Vidal-Suñé and Lopez-Panisello (2013), Sayed and Slimane (2014) and Rusu and Roman (2018) who empirically found a positive and statistically significant (at 1% level) relation between inflation rate and national competitiveness. Thus, an increase of the inflation rate determines an increase in business opportunities, because higher level of prices for products and services determines the increase of the expectations of entrepreneurs regarding potential earnings, but also stimulate business development and economic competitiveness. Although, the countries with high rates of inflation will not be between the countries with the highest competitiveness, they could register an improvement of competitiveness but will stay between the countries from the bottom of ranking, as shown by Kristjánsdóttir (2017).

The coefficient for tax rate emphasizes a negative and statistically significant (at 5% level) relation between total tax rate and economic competitiveness. Our result is similar with the ones of other studies (Summers 1988; Gray and Holtz-Eakin 2009; Knoll 2010; Miller and Kim 2008; Ecorys 2014) showing that high corporate tax rates can increase the administrative costs of the enterprises, reduce the profitability of the firm, reduce investments and labor productivity and implicitly reduce the global competitiveness of the economy. Therefore, reducing the total tax rates will result in attracting more investments, would stimulate enterprises productivity and will increase competitiveness of EU economies. In our times the economies are considered competitive, when they have reasonable corporate tax rates and low inflation (Kristjánsdóttir 2017).

A positive coefficient and statistically significant at 1% level was obtained for the control variable that measures the inflows of foreign direct investments. Thus, higher inflows of FDI for a country reduce unemployment rate, stimulate the enterprises to use modern techniques and technologies and to introduce new products, facilitate exports, leading to the development of local industries and thus, stimulating the economic competitiveness of that country. Our results are in agreement with the findings of Fontagné and Pajot (1997), Javorcik (2004), Ocharo and Musyoka (2018), Domazet and Marjanović (2018), which also emphasized the positive effects of increased inflows of foreign capital on economic competitiveness of countries.

Cost of business start-up procedures is another control variable which has a negative and statistically significant influence (at 1% level) on competitiveness. The negative coefficient shows that

higher costs for the creation of a new business reduce the competitiveness of the goods and services offered by the newly established firms and negatively affect the international competitiveness of the country because it makes it less attractive for foreign investors. The results are in agreement with previous studies (Iarossi 2009; Globerman and Georgopoulos 2012; Messaoud and Teheni 2014), which emphasized that more regulations about doing business in a country determine higher costs and lower competitiveness.

## 5. Conclusions

In this study we have investigated the effects of the factors measuring the quality of entrepreneurial activities on national competitiveness in the 28 European Union member states. As the review of literature showed, innovation rate and job creation are factors that significantly influence the level of economic development of countries and their national competitiveness. Besides these two main factors, we also have included in the analysis several control variables measuring the characteristics of the economic and business environment of the countries in the panel.

The purpose of our study was to test the hypotheses and to show the relationship between several indicators expressing the quality of entrepreneurship and of the economic environment on the level of competitiveness of European Union countries. As concluding remarks of our empirical investigation, we can affirm that the economic and business environment and the quality of entrepreneurial activity are key factors influencing national competitiveness for the European Union members. The quality of entrepreneurship is very important for the development of an economy. The innovative entrepreneurs are helping the development of markets and stimulate the increase of economic competitiveness. Our empirical results highlight that innovative entrepreneurial activities are positively and significantly related to national competitiveness of the European Union countries.

On the other hand, we have expected that increased employment to stimulate economic competitiveness of countries, but our results indicate an opposite relationship on short term between high job creation expectation rate and national competitiveness. This relationship appears because higher competitiveness is related to higher productivity, but increased productivity creates unemployment on short and medium terms, and employment on a long term.

The empirical results obtained show that the considered indicators are significantly influencing the competitiveness of the European Union countries and they are in accordance with the results of other empirical studies. Thus, innovation rate, inflation rate and FDI inflows are positively related with economic competitiveness of countries. High job creation expectation rate, tax rate and the costs of starting new business are negatively related with economic competitiveness of analzsed countries.

The added value of our study results from including in the analysis all 28 countries members of the European Union. Another plus of our study is the fact that we have considered the quality of entrepreneurship and its relationship with national competitiveness. There are only a few studies that analyze the impact of several aspects of entrepreneurship on the level of national competitiveness, but, to our knowledge, there are no studies testing the relation between the quality of entrepreneurial activities and national competitiveness. So, through this research we intended to fill this literature gap. Moreover, our study could be of interest to European policy makers who intend to enhance national competitiveness, because it points out the key role played by entrepreneurship for national economies, especially when it comes to its qualitative feature that has the potential of creating new jobs on a long term. Our results should draw the attention to the need of policy makers to identify and implement the best policies needed to sustain the increase of entrepreneurship quality in order to enhance competitiveness of European economies.

The limitations of our study come from the availability of the data and from the reduced number of the variables considered. This study represents a starting point of our research regarding the effects of entrepreneurial quality on economic competitiveness of countries. The quality of entrepreneurship is very important for the economic development of countries and for increasing their national competitiveness. Taking into account that the business sector, and especially the sector represented

by the small and medium enterprises is considered to be the engine of the economy, increasing its quality would have beneficial effects to the economy as a whole. So, the policy makers should focus on increasing the quality of entrepreneurial activity not only the numbers of entrepreneurs.

We intend to extend and develop our analysis in order to deepen the empirical investigation regarding the relationship between the quality of entrepreneurship and national competitiveness. Therefore, in further research we intent to add several explanatory variables measuring the quality of entrepreneurial activity which might influence the countries' economic and competitiveness development. Secondly, we intend to test if there are differences regarding the relationship between entrepreneurial quality and competitiveness when grouping the countries by their level of economic development, or by region. Several studies have shown that the impact of entrepreneurship on economic development is different according to the level of development of the country. For instance, in the case of developing countries there is no effect (Rusu and Tudose 2018) or a negative relation (Dvouletý et al. 2018) between the level of entrepreneurship and economic development and national competitiveness. Further research will test the relations mentioned above according to the level of economic development of the countries.

**Author Contributions:** V.D.R. and A.D. contributed equally to the elaboration of the paper. Both authors contributed to the elaboration of introduction, literature review, methodology, results and discussions and conclusions.

**Funding:** This work was supported by a grant of the "Alexandru Ioan Cuza" University of Iasi, within the Research Grants program, Grant UAIC, code GI-UAIC-2017-02.

**Conflicts of Interest:** The authors declare no conflict of interest.

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
