# Peer review of "The Quality of Entrepreneurial Activity and Economic Competitiveness in European Union Countries: A Panel Data Approach"

_admsci, doi:10.3390/admsci9020035_

Round 1

Reviewer 1 Report

Although the paper is well balanced in terms of the lit review, methodology and conclusions, one crucial aspect is missing which is related to why is this research important and how it contributes to advance the knowledge in this specific area? There is no mention whatsoever to the research gap that this paper addresses and why is it important. Moreover, in the literature review there is an appalling absence of ABS ranked journals in which an extensive list of published papers has addressed this theme, namely Entrepreneurship and Regional Development, Journal of Business Venturing and ETP. Clearly, supporting the lit review in book chapters, books and papers published in mostly irrelevant journals in this field is not enough to position a paper that aims at contributing, at least minimally, to advance the knowledge in this important area.

Author Response

Dear Reviewer,

We would like to thank you for the suggestions, recommendations and comments in relation with the scientific approach in our paper. We considered carefully each one of your suggestions. All of them are very relevant for us and represented a reliable guide and strong support in our efforts to increase the quality of our paper. 

Starting from your specific suggestions (in bold), the changes introduced in paper are as follows:

Although the paper is well balanced in terms of the lit review, methodology and conclusions, one crucial aspect is missing which is related to why is this research important and how it contributes to advance the knowledge in this specific area? There is no mention whatsoever to the research gap that this paper addresses and why is it important. 

Response: We have presented why is our research important and which is the gap that it address (Please see line 37-41 and 521-533). In order to be more clear we have added several explications also in the introduction section (Please see line: 51-61)

Moreover, in the literature review there is an appalling absence of ABS ranked journals in which an extensive list of published papers has addressed this theme, namely Entrepreneurship and Regional Development, Journal of Business Venturing and ETP. Clearly, supporting the lit review in book chapters, books and papers published in mostly irrelevant journals in this field is not enough to position a paper that aims at contributing, at least minimally, to advance the knowledge in this important area.

Response: We have take into account the suggestion and we have updated the Literature review section accordingly.

Sincerely yours,

The authors

Reviewer 2 Report

The paper analyzes the relationship between the competitiveness of the regions and the quality of entrepreneurial activity. The quality of the entrepreneurial activity is measured through the percentage of enterprises that indicate having a new product for some consumers and the expectations of generating employment in the long term. Both variables are a percentage of those involved in Total early stage entrepreneurial activity.

However, some considerations are necessary for the correct understanding of the article:
In the introduction it is indicated that the period of analysis is the period 2011-2017 but the reason of this period is not justified. Bearing in mind that it is a phase conditioned by the deepening of the crisis and the progressive exit of it. In addition, the effects of the same have not occurred with the same intensity in all EU countries. It also indicates the existence of some control variables to measure the influence of the environment but it is not justified why these variables have been considered and not others. In this section it would be advisable to raise the environment and the motivations that lead to the generation of research. That is, to define the economic, social and political context.

In the literature review, the article focuses on the different existing concepts on competitiveness and comments on the main conclusions of some studies. However, a more specific revision is required. Especially on empirical evidence within the euro zone and articles that relate the phenomenon of competitiveness and entrepreneurial activity. Many of the references that appear in the results section should appear in this review. In the review, the effects of entrepreneurs who innovate with economic growth should be highlighted.

In the section on sources and methods, the dependent variable (GCI) should be explained in more detail, as well as the GEM and the measurement of TEA. It is hardly indicated what information this variable collects and the different types of entrepreneurship. It should be noted that the choice of independent variables is based on estimates or perceptions of the entrepreneurs. The same with the data provided by the World Bank. The reason for the period and the choice of variables must be justified. It must be explained why there is a different number of observations in some of the variables. The choice of methodology is appropriate, but the choice of models should be reduced.

Regarding the results part, the sign and the significance of the variables are hardly commented. They should take advantage of the available data and explain both the evolution of some of the variables and the different behaviors that may occur in the EU
countries. However, when proceeding with the model and explaining what should
be considered:
a) The percentage of early TEA is used, which indicates that its product is new, but a relativization of the variable does not appear. That is, there is no variable that weighs the importance of entrepreneurship within each region.

b) It is indicated that the article coincides with other studies when considering innovation as a determinant of competitiveness, but with the data that can not be justified. Thinking that you have a new product or that is not offered by other companies does not necessarily imply innovation. In addition, one of the pillars of the
GCI is innovation.

c) There is no clear appreciation of the relationship between entrepreneurship, job creation and competitiveness.

d) The control variables should be more clearly related to those associated with innovation and employment.

Finally, the conclusions should indicate the limitations of the study and offer some recommendations from the point of view of economic policies.

Author Response

 Dear Reviewer,

We would like to thank you for the suggestions, recommendations and comments in relation with the scientific approach in our paper. We considered carefully each one of your suggestions. All of them are very relevant for us and represented a reliable guide and strong support in our efforts to increase the quality of our paper. 

Starting from your specific suggestions (in bold), the changes introduced in paper are as follows:

The paper analyzes the relationship between the competitiveness of the regions and the quality of entrepreneurial activity. The quality of the entrepreneurial activity is measured through the percentage of enterprises that indicate having a new product for some consumers and the expectations of generating employment in the long term. Both variables are a percentage of those involved in Total early stage entrepreneurial activity.

However, some considerations are necessary for the correct understanding of the article:

In the introduction it is indicated that the period of analysis is the period 2011-2017 but the reason of this period is not justified. Bearing in mind that it is a phase conditioned by the deepening of the crisis and the progressive exit of it. In addition, the effects of the same have not occurred with the same intensity in all EU countries. 

Response: we have added the motivation for choosing the period, the availability of data. We also added some explanations regarding the fact that the period considered includes the years after the recent financial crisis and that our results might be influenced by this situation (Please see line: 245-253)

It also indicates the existence of some control variables to measure the influence of the environment but it is not justified why these variables have been considered and not others. In this section it would be advisable to raise the environment and the motivations that lead to the generation of research. That is, to define the economic, social and political context. 

Response: we have added several explanations why we have chosen the control variables, their relation with national competitiveness and entrepreneurship (line: 406-433).

In the literature review, the article focuses on the different existing concepts on competitiveness and comments on the main conclusions of some studies. However, a more specific revision is required. Especially on empirical evidence within the euro zone and articles that relate the phenomenon of competitiveness and entrepreneurial activity. Many of the references that appear in the results section should appear in this review. 

Response: We consider better to mantain the literature review more generrally , as we have seen also in others papers, and the references mentioned in the results section to keep them as they are because we avoid repetition and also help us to highlight better our results compared to the results of ohter studies

In the review, the effects of entrepreneurs who innovate with economic growth should be highlighted.

Response: is mentioned at rows 26-29 of the paper. We have also added at rows 131-134 „In their study, González-Pernía et al. (2015) highlighted the importance of innovative entrepreneurs, even they represent a small portion of the entire population of business founders, they have an extraordinary economic impact, as they develop new technologies, create new jobs and enhance the revitalization capacity of territories.”

In the section on sources and methods, the dependent variable (GCI) should be explained in more detail, as well as the GEM and the measurement of TEA. It is hardly indicated what information this variable collects and the different types of entrepreneurship. It should be noted that the choice of independent variables is based on estimates or perceptions of the entrepreneurs. The same with the data provided by the World Bank. The reason for the period and the choice of variables must be justified. It must be explained why there is a different number of observations in some of the variables. The choice of methodology is appropriate, but the choice of models should be reduced.

Response: Please see line 284-293 for explanations regarding the variable GCI. We have added in the table explanations for the measurement of TEA, We also added explanations why we have obtained different number of observations for some variables

Regarding the results part, the sign and the significance of the variables are hardly commented. They should take advantage of the available data and explain both the evolution of some of the variables and the different behaviors that may occur in the EU
countries. However, when proceeding with the model and explaining what should be considered:
a) The percentage of early TEA is used, which indicates that its product is new, but a relativization of the variable does not appear. That is, there is no variable that weighs the importance of entrepreneurship within each region. 
b) It is indicated that the article coincides with other studies when considering innovation as a determinant of competitiveness, but with the data that cannot be justified. Thinking that you have a new product or that is not offered by other companies does not necessarily imply innovation. In addition, one of the pillars of the
GCI is innovation.
c) There is no clear appreciation of the relationship between entrepreneurship, job creation and competitiveness.
d) The control variables should be more clearly related to those associated with innovation and employment.

Response: We added several explanations regarding the significant of the coefficients obtained. We have added explanations why choosing the control variables, and also have described their relationship with the other variables considered in the analysis (Please see line 405-432). Global Entrepreneurship Monitor considers the variables innovation (which we have considered in our analysis) as an indicator measuring the level of innovation of entrepreneurs. Thus we have adopted it to be representative for the purpose of our empirical analysis. Considering those two variables (innovation and high job creation expectation rate) as variables for measuring the quality of entrepreneurship is the novelty element of our paper. This is an incipient study on this problem, and, based on the results obtained here in further research we intent to extend the analysis by considering more variables. 

Finally, the conclusions should indicate the limitations of the study and offer some recommendations from the point of view of economic policies.

Response: we added the limitations of our study, and some recommendations.

Sincerely yours,

The authors

Round 2

Reviewer 1 Report

I think that you made a good attempt to improve the manuscript. However, the literature review is still a bit dense. I understand that you want to bring an in-depth analysis of the relevant literature to support your research, but would try to reduce it a bit and focus on the most important aspects by summarizing the least important. Maybe a table representation of the literature would help to the flow of the reading instead of just text? Anyway this is a suggestion for improvement.

I reckon that in the methods section you bring in new literature which has not been addressed before, so I think that this needs to be tied up a bit as well.

Some important aspects that need to be corrected are the numbering of the sections. There is no section 4 and sections 3 and 5 have the same title. I also recommend a thorough reading of the manuscript preferably by a native english speaker to edit and improve the overall reading experience.  

Last but not least, some quotes in the text are not properly referenced.

Author Response

Dear Reviewer,

We would like to thank you for the suggestions, recommendations and comments in relation with the scientific approach in our paper. We considered carefully each one of your suggestions. All of them are very relevant for us and represented a reliable guide and strong support in our efforts to increase the quality of our paper. 

Starting from your specific suggestions (in bold), the changes introduced in paper are as follows:

I think that you made a good attempt to improve the manuscript. However, the literature review is still a bit dense. I understand that you want to bring an in-depth analysis of the relevant literature to support your research, but would try to reduce it a bit and focus on the most important aspects by summarizing the least important. Maybe a table representation of the literature would help to the flow of the reading instead of just text? Anyway this is a suggestion for improvement.

Response: We added Table 1 in Literature review section.

I reckon that in the methods section you bring in new literature which has not been addressed before, so I think that this needs to be tied up a bit as well.

Response: Table 1 brings in the literature review section, the papers mentioned in method section.

Some important aspects that need to be corrected are the numbering of the sections. There is no section 4 and sections 3 and 5 have the same title. I also recommend a thorough reading of the manuscript preferably by a native english speaker to edit and improve the overall reading experience.  

Response: we have corrected the numbering of the sections. We also gave the manuscript to be read by a native English speaker and made some changes.

Last but not least, some quotes in the text are not properly referenced.

Response: we have corrected some reference in the text

Sincerely yours,

The authors

Reviewer 2 Report

The authors must reinforce the bibliographic review, highlighting studies that consider all the countries of the European Union. It is indicated that there are still few investigations that inquire about the relationship between competitiveness and quality of the entrepreneurial activity in terms of innovation. It is recommended to quote them and present the main results of the same in order to quantify the contribution of the paper. The review should be more specific in relation to the topics presented in the introduction, with articles that analyze the TEA of recent initiatives. It is recommended to be more selective in the literature review, eliminating those references that have to do with developing countries.

In the part of materials and methods, it is repeated twice (point 3 and point 5). A graphic model that represents the model and the relationship between the variables would be more illustrative. It requires a greater justification of how the chosen variables allow establishing a relationship between entrepreneurship and competitiveness.

The part of results becomes necessary a deepening in them, beyond the interpretation of the sign and the significance. The authors should try to identify some of the differences between the European regions and not present such aggregate results.

Author Response

 Dear Reviewer,

We would like to thank you for the suggestions, recommendations and comments in relation with the scientific approach in our paper. We considered carefully each one of your suggestions. All of them are very relevant for us and represented a reliable guide and strong support in our efforts to increase the quality of our paper. 

Starting from your specific suggestions (in bold), the changes introduced in paper are as follows:

The authors must reinforce the bibliographic review, highlighting studies that consider all the countries of the European Union. It is indicated that there are still few investigations that inquire about the relationship between competitiveness and quality of the entrepreneurial activity in terms of innovation. It is recommended to quote them and present the main results of the same in order to quantify the contribution of the paper. The review should be more specific in relation to the topics presented in the introduction, with articles that analyze the TEA of recent initiatives. It is recommended to be more selective in the literature review, eliminating those references that have to do with developing countries.

Response: The influence of qualitative entrepreneurship, measured by innovation and job creation, on national competitiveness has been poorly discussed in the literature. Even so, we have mentioned the research of Bosma et al. 2018, Ciocanel and Pavelescu 2015, Kritikos 2014, Szabo and Herman 2012. EU is formed by developed countries and also developing countries . This is the reason why we also mentioned in the review papers which have analyzed the relationship between entrepreneurship and competitiveness in developing countries.

In the part of materials and methods, it is repeated twice (point 3 and point 5). A graphic model that represents the model and the relationship between the variables would be more illustrative. It requires a greater justification of how the chosen variables allow establishing a relationship between entrepreneurship and competitiveness.

Response: we changed the numbers and names of the sections mentioned. We added a graphical representation of the relationship between variables.

The part of results becomes necessary a deepening in them, beyond the interpretation of the sign and the significance. The authors should try to identify some of the differences between the European regions and not present such aggregate results.

Response: thank you for the suggestions. As we have mentioned in the conclusion section this is an incipient study, a starting point for our preoccupations regarding the relationship between the quality of entrepreneurship and national competitiveness and we considered it in a more general way, for all the European Union member countries, but, in further research we intend to extend our empirical investigation and to consider grouping the EU countries either by their level of economic development, or by region, or both. That is why our results are interpreted for the EU as a whole.

Sincerely yours,

The authors

Round 3

Reviewer 2 Report

Checked the changes made in the test.